

# The triglycerides-glucose index and the triglycerides to high-density lipoprotein cholesterol ratio are both effective predictors of in-hospital death in non-diabetic patients with AMI

Jiaqi Guo[1], Zhenjun Ji[1], Abdlay Carvalho[1], Linglin Qian[1], Jingjing Ji[1], Yu Jiang[1], Guiren Liu[2], Genshan Ma[1] and Yuyu Yao[1]

[1] Department of Cardiology, Zhongda Hospital, School of Medicine, Southeast University, Nanjing, Jiangsu, China
[2] Department of Epidemiology and Health Statistics, School of Public Health, Southeast University, Nanjing, Jiangsu, China

Corresponding author
Yuyu Yao, yaoyuyunj@hotmail.com

## ABSTRACT

**Background**. The triglycerides-glucose index (TyG) and the triglycerides to high-density lipoprotein cholesterol (TG/HDL-C) are simple indicators for assessing insulin resistance in epidemiological studies. We aimed to clarify the relationship between indicators of insulin resistance and prognosis in non-diabetic acute myocardial infarction (AMI) patients.

**Methods**. A total of 1,648 AMI patients without diabetes were enrolled from the Department of Cardiology, Zhongda Hospital, between 2012.03 and 2018.12. The medical history, laboratory and imaging data of patients were collected through the medical record system, and all-cause death events were recorded. Pearson analysis was used to study the correlation among different variables. Logistic regression analysis was used to analyze the predictive effect of TyG and TG/HDL-C in in-hospital death of AMI patients.

**Results**. 1. In AMI group, the TyG index was significantly increased in death groups compared to no-death groups ($P = 0.025$). TG/HDL-C was not significantly increased in the death group of AMI patients ($P = 0.588$). The patients were respectively divided into Q1-Q4 groups and T1-T4 groups according to the quartiles of TyG and TG/HDL-C. The trends of in-hospital mortality in the Q4 group of TyG and T4 group of TG/HDL-C were higher than in other groups, although these differences were not significant. 2. Pearson correlation analysis showed that TyG was positively correlated with lipid-related markers, including ApoB ($r = 0.248$, $P < 0.001$), total cholesterol (TC) ($r = 0.270$, $P < 0.001$), low-density lipoprotein cholesterol (LDL-C) ($r = 0.238$, $P < 0.001$). Spearman analysis showed that TG/HDL-C was also positively associated with TC ($r = 0.107$, $P < 0.001$), ApoB ($r = 0.180$, $P < 0.001$) and LDL-C ($r = 0.164$, $P < 0.001$). 3. Logistic regression analysis showed that TyG (OR = 3.106, 95% CI [2.122–4.547], $P < 0.001$) and TG/HDL-C (OR = 1.167, 95% CI [1.062–1.282], $P = 0.001$) were both important factors to predict the in-hospital death of AMI patients without diabetes.

**Conclusions**. TyG index and TG/HDL-C, as emerged simple markers of insulin resistance, were both important predictors of in-hospital death in AMI patients without diabetes.

## INTRODUCTION

Acute coronary syndrome (ACS) is a severe type of coronary heart disease (*Reed, Rossi & Cannon, 2017*). The Global Registry of acute coronary events (GRACE) study showed that the 1-year mortality of ACS patients was about 15%, and the cumulative 5-year mortality was about 20% (*Fox et al., 2010*). The mortality of acute myocardial infarction (AMI) is still on the rise in China. Diabetes and metabolic syndrome are important risk factors for AMI (*DeFilippis et al., 2019*). Identifying high-risk patients, early diagnosis and timely intervention of patients with AMI to reduce mortality and improve prognosis have been important research directions.

Insulin resistance is a significant risk factor for cardiovascular disease (CVD), which is associated with decreased cardiac autonomic function in the elderly without diabetes (*Laakso & Kuusisto, 2014*; *Ormazabal et al., 2018*; *Tenenbaum et al., 2007*; *Howard et al., 1996*). Insulin resistance is characterized by the loss of endogenous or exogenous insulin effect, resulting in glucose uptake and utilization disorders (*James, Stockli & Birnbaum, 2021*). A study showed that nearly 58.4% of non-diabetic Parkinson's patients have been diagnosed with insulin resistance (*Hogg et al., 2018*). Tissue-specific insulin resistance, such as abnormalities in the liver, muscle and adipose tissue, can lead to endothelial dysfunction and macrophage accumulation, causing severe CVD events (*Poon et al., 2020*; *Jia et al., 2021*; *Ryu et al., 2021*).

The triglycerides-glucose index (TyG) and the triglycerides to high-density lipoprotein cholesterol (TG/HDL-C) are simple indicators for assessing insulin resistance in epidemiological studies. TyG is the product of fasting triglyceride and fasting blood glucose (*Wang et al., 2021a*; *Wang et al., 2021b*). There is a positive correlation between TyG index and insulin resistance (*Dikaiakou et al., 2020*). Compared with the homeostasis model assessment of insulin resistance (HOMA-IR) index and the euglycemic hyperinsulinemic clamp test, TyG is a new, concise and reliable alternative index for evaluating insulin resistance (*Wang et al., 2021a*; *Wang et al., 2021b*). Moreover, studies have shown that a higher TyG index is associated with an increased risk of type 2 diabetes (*Xuan et al., 2021*; *Dikaiakou et al., 2020*). In addition, TyG index is positively correlated with coronary artery calcification and is an effective indicator for predicting the progress of CVD (*Park et al., 2019*).

TG/HDL-C is also a feasible indicator for evaluating insulin resistance (*Zhou et al., 2016*). A study that enrolled 9764 adults (average age 56 years old) who had participated in the REACTION study in a Guangzhou community found that, compared with other blood lipid markers, TG/HDL-C may be a better indicator for assessing insulin resistance and diabetes among middle-aged and elderly Chinese populations (*Lin et al., 2018*). *Zare et al. (2022)* used TG/HDL-C as a surrogate index of insulin resistance in people with pulmonary hypertension without a history of diabetes, and defined TG/HDL-C>3.0 as an

indicator of insulin resistance. This study found that insulin resistance was not associated with all-cause mortality in pulmonary hypertension (*Zare et al., 2022*; *Naderi et al., 2014*).

Due to the crucial roles of TyG and TG/HDL-C in insulin resistance, this study aimed to use TyG and TG/HDL-C to explore the effects of insulin resistance on cardiac function and in-hospital death in non-diabetic AMI patients, and to provide guidance for the stratification of clinical risk in patients with AMI.

## MATERIALS & METHODS

### Population

This design was an observational and retrospective study. 1648 AMI patients without diabetes, consisting of 774 NSTEMI and 874 STEMI patients, were enrolled from the Department of Cardiology, Zhongda Hospital between 2012.03 and 2018.12. Inclusion criteria: (1) Age >18 years old; (2) The patient was diagnosed with AMI for the first time. Exclusion criteria: (1) Pregnant or lactating women; (2) Patients with severe diseases, such as advanced malignant tumors, and patients with an expected survival time of less than 3 months; (3) Diabetic patients. The Ethics Committee of Zhongda Hospital affiliated to Southeast University (2020ZDSYLL164-P01) approved this study and exempted the need for informed consent signature.

### Data collection

The demographic characteristics, medical history, laboratory and imaging data of patients were collected by the medical record system (Yidu Cloud System, China), and death events were recorded. In-hospital death was defined as all-cause death due to cardiovascular and non-cardiovascular disease, and all deaths were reported with medical records (*Yusuf et al., 2020*; *Lee et al., 2019*). The demographic data included age, gender, smoking history and systolic blood pressure (SBP). The medical history included history of hypertension (HBP) and diabetes. Blood routine examination included white blood cells (WBC), neutrophils, lymphocytes, platelet (PLT) and hemoglobin (Hb). The biochemical test included alanine aminotransferase (ALT), lactate dehydrogenase (LDH), aspartate aminotransferase (AST), blood urea nitrogen (BUN), uric acid (UA), total cholesterol (TC), triglyceride (TG), albumin (ALB), glucose (GLU), high-sensitivity C-reactive protein (hsCRP), low-density lipoprotein cholesterol (LDL-C) and high-density lipoprotein cholesterol (HDL-C). Left ventricular ejection fraction (LVEF) data were collected from echocardiography were collected. Fasting blood glucose and lipid profiles for all participants were assessed upon admission. The TyG index was calculated by ln(TG [mg/dL]*GLU [mg/dL]/2) (*Park et al., 2019*). TG/HDL-C was calculated as TG dividing by HDL-C (*Wu et al., 2021*).

### Statistics

Data were analyzed by SPSS 23.0 and GraphPad Prism 8 software. The Shapiro–Wilk test and Q-Q plots were both used to test normality of continuous variables. For continuous data conforming to a normal distribution, independent-sample $T$-test was used for comparison between two groups. If the data did not follow a normal distribution, Mann–Whitney U was used for comparison among groups. For categorical variables, comparison was

performed by Chi-square test or Fisher's exact test. For variables conforming to normality, Pearson analysis were used to test the correlation between TyG or TG/HDL-C and other variables. For rank data or data that do not conform to a normal distribution, Spearman was used for correlation analysis. Univariate and multivariate logistic regression analyses were used to analyze the predictive effect of TyG or TG/HDL-C in in-hospital death of AMI patients. The variables included in logistic regression must conform to independence and linearity.

## RESULTS

### Basic characteristics of enrolled patients

The Shapiro–Wilk test and Q-Q plots were both used to test the normality of continuous variables (Supplemental Materials 1 and 2). Homogeneity test for variance was performed (Supplemental Materials 3). Continuous variables with normality included age, SBP, neutrophils, lymphocytes, PLT, Hb, LDL-C, HDL-C, UA, TC, ALB and LVEF. Two-sample independent t-tests were used for comparison between death and no-death groups. Continuous variables that did not conform to normality included WBC, ALT, AST, LDH, BUN, cTnI, Creatinine, GLU, hsCRP and HbA1c. Rank sum test was used. The Chi-square test was used to compare categorical variables, including sex, smoking, medical history, and concomitant medication. In the STEMI and NSTEMI groups, there were no differences in sex and hypertension history between the death and no-death groups ($P > 0.05$). The age in the death groups was higher than in the no-death groups ($P < 0.001$). The WBC count and neutrophils were also significantly higher than that of no-death groups ($P < 0.05$), indicating a severe inflammatory state in death groups. There were no differences in ALT and AST between the two groups ($P > 0.05$). The ALB levels were significantly lower in death groups than in no-death groups ($P < 0.001$). The creatinine level was higher in death groups than in no-death groups ($P < 0.01$) (Table 1).

### TyG and TG/HDL-C, both indicators of insulin resistance, are closely related to death after myocardial infarction

Normality of TyG was tested by the Shapiro–Wilk test and Q-Q plots. The absolute value of the sample kurtosis is less than 10 and the absolute value of the skewness is less than 3, TyG can be described as basically conforming to the normal distribution. Independent-sample t-test was used for comparison of TyG. The TyG index was significantly increased in the death groups compared to the no-death groups in AMI ($8.86 \pm 0.59$ vs $9.22 \pm 0.97$, $P = 0.001$), STEMI ($8.90 \pm 0.60$ vs $9.24 \pm 0.95$, $P = 0.024$) and NSTEMI ($8.81 \pm 0.58$ vs $9.19 \pm 1.00$, $P = 0.022$) patients (Fig. 1A). Mann–Whitney U method was used for comparison of TG/HDL-C not conforming to normality. TG/HDL-C was not significantly increased in the death group of AMI patients ($P = 0.588$) (Fig. 1B). The patients were also divided into different groups according to the quartiles of TyG (Q1 <8.4543, Q2 8.4543−8.8418, Q3 8.8418−9.2294, Q4 ≥9.2294) and TG/HDL-C (T1 <2.0270, T2 2.0270−2.9487, T3 2.9487−4.3456, T4 ≥4.3456). The Chi-square test was used to test for differences in the trend of in-hospital mortality. The results showed that in-hospital mortality in the Q4 group of TyG (6.80%) and T4 group of TG/HDL-C (5.76%) was higher than in other

Guo et al. (2022), *PeerJ*, DOI 10.7717/peerj.14346

**Table 1  Basic characteristics of AMI patients.** For continuous data conformed to normality, data was described as mean ± standard deviation (X ± SD). For data did not conform to normality, data was described by quartiles [50% (25%, 75%)].

| Characteristics | STEMI | | P value | NSTEMI | | P value |
|---|---|---|---|---|---|---|
| | Without death (n = 829) | Death (n = 45) | | Without death (n = 734) | Death (n = 40) | |
| Demographic characteristics | | | | | | |
| Sex (female) | 175 (21.11%) | 10 (22.22%) | 0.859 | 225 (30.65%) | 18 (45.00%) | 0.057 |
| Age (years) | 67.65 ± 14.01 | 78.87 ± 14.40 | <0.001[***] | 73.37 ± 13.21 | 85.90 ± 6.38 | <0.001[***] |
| Smoking | 426 (51.39%) | 18 (40.00%) | 0.137 | 280 (38.15%) | 9 (22.50%) | <0.046[*] |
| SBP (mmHg) | 127.67 ± 21.37 | 121.23 ± 25.80 | 0.054 | 134.87 ± 22.18 | 122.87 ± 27.51 | 0.001[**] |
| Blood routine examination | | | | | | |
| Neutrophils ($10^9$/L) | 8.19 ± 4.38 | 10.56 ± 6.47 | 0.022[*] | 6.05 ± 3.45 | 8.38 ± 5.67 | 0.014[*] |
| Lymphocytes ($10^9$/L) | 1.39 ± 0.66 | 1.34 ± 0.90 | 0.620 | 1.45 ± 0.69 | 1.14 ± 0.73 | 0.006[**] |
| WBC ($10^9$/L) | 9.34 (7.08, 12.19) | 11.21 (7.14, 16.93) | 0.043[*] | 7.32 (6.00, 9.28) | 8.60 (6.49, 11.76) | <0.001[***] |
| PLT ($10^9$/L) | 207.79 ± 64.95 | 195.53 ± 93.16 | 0.399 | 194.60 ± 70.01 | 178.38 ± 80.78 | 0.158 |
| Hb (g/L) | 133.31 ± 20.11 | 119.72 ± 26.36 | 0.002[**] | 130.76 ± 22.32 | 110.18 ± 26.49 | <0.001[***] |
| Biochemical test | | | | | | |
| ALT (U/L) | 39.00 (24.00, 61.00) | 34.00 (15.50, 84.50) | 0.655 | 24.00 (16.00, 35.00) | 30.00 (13.00, 38.75) | 0.223 |
| AST (U/L) | 98.00 (36.00, 207.75) | 57.00 (28.00, 306.50) | 0.483 | 33.00 (22.00, 62.00) | 47.50 (21.25, 150.25) | 0.057 |
| LDH (U/L) | 426.50 (251.25, 737.50) | 405.00 (272.50,877.50) | 0.407 | 235.00 (178.00, 311.00) | 300.00 (195.75, 546.50) | 0.010[*] |
| LDL-C (mmol/L) | 2.80 ± 0.86 | 2.51 ± 1.06 | 0.084 | 2.71 ± 0.91 | 2.31 ± 0.90 | 0.008[**] |
| HDL-C (mmol/L) | 1.10 ± 0.26 | 0.98 ± 0.26 | 0.007[**] | 1.09 ± 0.27 | 1.00 ± 0.34 | 0.106 |
| BUN (mmol/L) | 5.10 (4.20, 6.70) | 8.80 (6.05, 15.50) | <0.001[***] | 5.90 (4.70, 7.90) | 10.35 (8.18, 15.08) | <0.001[***] |
| UA (μmol/L) | 349.80 ± 111.27 | 434.98 ± 211.36 | 0.010[*] | 362.79 ± 121.54 | 385.80 ± 157.95 | 0.370 |
| cTnI (ng/ml) | 3.82 (0.57, 12.00) | 2.84 (0.11, 11.50) | 0.219 | 0.67 (0.13, 2.92) | 1.60 (0.15, 7.00) | 0.165 |
| TC (mmol/L) | 4.48 ± 1.11 | 4.09 ± 1.52 | 0.099 | 4.39 ± 1.16 | 3.94 ± 1.23 | 0.017[*] |
| ALB (g/L) | 37.56 ± 4.65 | 32.43 ± 6.58 | <0.001[***] | 37.23 ± 4.96 | 32.41 ± 6.64 | <0.001[***] |
| Creatinine (μmol/L) | 80.00 (66.00, 96.50) | 119.00 (91.50, 191.00) | <0.001[***] | 86.50 (69.00, 109.00) | 128.50 (103.50,212.75) | 0.009[**] |
| GLU (mmol/L) | 6.41 (5.53, 7.62) | 7.56 (5.44, 9.41) | 0.027[*] | 6.13 (5.37, 7.19) | 6.68 (5.61, 8.29) | 0.042[*] |
| hsCRP (mg/L) | 16.35 (4.27, 49.98) | 24.50 (12.00, 95.90) | 0.083 | 9.56 (2.33, 37.38) | 44.10 (12.18, 81.45) | 0.002[**] |
| HbA1c (%) | 5.70 (5.40, 6.30) | 6.10 (5.95, 6.90) | 0.081 | 5.80 (5.50, 6.20) | 6.25 (5.85, 6.88) | 0.105 |
| Echocardiography | | | | | | |
| LVEF | 0.55 ± 0.12 | 0.49 ± 0.17 | 0.093 | 0.60 ± 0.13 | 0.51 ± 0.17 | 0.001[**] |
| Medical history | | | | | | |
| HBP | 464 (55.97%) | 24 (53.33%) | 0.729 | 522 (71.12%) | 32 (80.00%) | 0.225 |
| CKD | 30 (3.61%) | 6 (13.33%) | 0.001[**] | 57 (7.77%) | 8 (22.00%) | 0.007[**] |

Guo et al. (2022), *PeerJ*, DOI 10.7717/peerj.14346

**Table 1** (*continued*)

| Characteristics | STEMI | | P value | NSTEMI | | P value |
|---|---|---|---|---|---|---|
| | Without death ($n = 829$) | Death ($n = 45$) | | Without death ($n = 734$) | Death ($n = 40$) | |
| Concomitant medication | | | | | | |
| Hypotensive drugs | 601 (72.50%) | 11 (24.44%) | <0.001[***] | 524 (71.39%) | 16 (40.00%) | <0.001[***] |
| Lipid-lowering drugs | 783 (94.45%) | 34 (75.56%) | <0.001[***] | 704 (95.91%) | 28 (70.00%) | <0.001[***] |
| Aspirin | 787 (94.93%) | 33 (73.33%) | <0.001[***] | 710 (96.73%) | 28 (70.00%) | <0.001[***] |
| Diuretics | 333 (40.17%) | 10 (22.22%) | 0.016[*] | 280 (38.15%) | 8 (20.00%) | 0.021[*] |
| $\beta$-blockers | 704 (84.92%) | 20 (44.44%) | <0.001[***] | 605 (82.43%) | 20 (50.00%) | <0.001[***] |

**Notes.**

[*]$P < 0.05$

[**]$P < 0.01$

[***]$P < 0.001$

SBP, Systolic blood pressure; LDH, Lactate dehydrogenase; ALT, Alanine aminotransferase; AST, Aspartate aminotransferase; LDL-C, Low-density lipoprotein cholesterol; HDL-C, High-density lipoprotein cholesterol; BUN, Blood urea nitrogen; UA, Uric acid; TC, Total cholesterol; ALB, Albumin; GLU, Glucose; WBC, White blood cells; Hb, Hemoglobin; HbA1C, Glycosylated hemoglobin; LVEF, Left ventricular ejected fraction; HBP, Hypertension; CKD, Chronic kidney disease; PLT, Platelet.

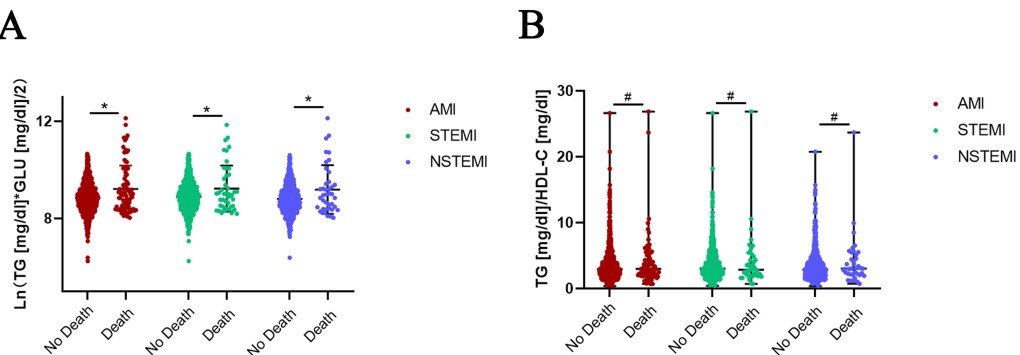

**Figure 1 Expression of TyG and TG/HDL-C in no-death and death groups of AMI, STEMI and NSTEMI.** (A) Independent-sample *T*-test was used for comparison of TyG conforming to normality. The TyG index was significantly increased in death groups compared to no-death groups in AMI (8.86 ± 0.59 *vs* 9.22 ± 0.97, *P* = 0.001), STEMI (8.90 ± 0.60 *vs* 9.24 ± 0.95, *P* = 0.024) and NSTEMI (8.81 ± 0.58 *vs* 9.19 ± 1.00, *P* = 0.022) patients. (B) Mann–Whitney U method was used for comparison of TG/HDL-C not conforming to normality. TG/HDL-C was not significantly increased in the death group of AMI patients (*P* = 0.588), as well as STEMI (*P* = 0.961) and NSTEMI (*P* = 0.456) subgroups. Data was presented by mean with SD (TyG) and median with range (TG/HDL-C). * *P* < 0.05, # *P* > 0.05.

groups (Q1 = 5.10%, Q2 = 3.64%, Q3 = 5.10%; T1 = 4.51%, T2 = 4.75%, T3 = 4.74%), despite the lack of statistically significant differences among these groups.

## Pearson analysis of TyG and TG/HDL-C

Pearson was used for correlation analysis for the variables not conforming to normality, otherwise, Spearman was used. Correlation analysis showed that TyG was positively correlated with WBC ($r = 0.113$, $P < 0.001$), neutrophils ($r = 0.090$, $P < 0.001$, Hb ($r = 0.183$, $P < 0.001$), UA ($r = 0.113$, $P < 0.001$), TC ($r = 0.270$, $P < 0.001$), LDL-C ($r = 0.238$, $P < 0.001$), ApoB ($r = 0.248$, $P < 0.001$) and ALB ($r = 0.169$, $P < 0.001$). TyG was negatively associated with age ($r = -0.217$, $P < 0.001$) and BUN ($r = -0.077$, $P = 0.002$), and not related to SBP ($r = 0.036$, $P = 0.147$) (Table 2). TG/HDL-C was positively associated with lymphocytes ($r = 0.240$, $P < 0.001$), Hb ($r = 0.227$, $P < 0.001$), ALB ($r = 0.165$, $P < 0.001$), TC ($r = 0.107$, $P < 0.001$), UA ($r = 0.185$, $P < 0.001$), ApoB ($r = 0.180$, $P < 0.001$), LDL-C ($r = 0.164$, $P < 0.001$) and LVEF ($r = 0.090$, $P = 0.001$), and negatively related to age ($r = -0.326$, $P < 0.001$), BUN ($r = -0.080$, $P = 0.001$), hsCRP ($r = -0.103$, $P = 0.024$) and ApoA ($r = -0.311$, $P < 0.001$) (Table 2).

## TyG and TG/HDL-C are important independent predictors of in-hospital death in AMI

Since this study only recorded the death status of enrolled subjects without recording the time of death, logistic regression instead of Cox regression was used to study the predictive ability of TyG and TG/HDL-C on in-hospital death of AMI patients. Due to a large number of variables (about 100) in this study and the sample size of 1,648 patients, the inclusion of excess variables into the logistic model could affect the model's accuracy. Therefore, we chose to perform univariate logistic regression to screen out the significant variables that were subsequently included in multivariate logistic regression without

**Table 2  Correlations analysis about TyG and TG/HDL-C.** For variables conforming to normality, Pearson was used for correlation analysis, otherwise, Spearman was used. Correlation analysis among age, SBP, WBC, neutrophils, Hb, ALB, TC, UA, hsCRP, ApoB, ApoA, LDL-C, LVEF and TyG were performed by Pearson. Correlation analysis among lymphocytes, creatinine, BUN and TyG were performed by Spearman. Correlation analysis between TG/HDL-C and other factors were performed by Spearman.

| Variables | Correlation analysis of TyG | | Correlation analysis of TG/HDL-C | |
|---|---|---|---|---|
| | r | p | r | p |
| Age (years) | −0.217 | <0.001[***] | −0.326 | <0.001[***] |
| SBP (mmHg) | 0.036 | 0.147 | 0.007 | 0.794 |
| WBC ($10^9$/L) | 0.113 | <0.001[***] | 0.034 | 0.192 |
| Neutrophils ($10^9$/L) | 0.090 | <0.001[***] | −0.018 | 0.488 |
| Lymphocytes ($10^9$/L) | 0.172 | <0.001[***] | 0.240 | <0.001[***] |
| Hb (g/L) | 0.183 | <0.001[***] | 0.227 | <0.001[***] |
| ALB (g/L) | 0.169 | <0.001[***] | 0.165 | <0.001[***] |
| Creatinine (μmol/L) | −0.029 | 0.238 | 0.009 | 0.706 |
| TC (mmol/L) | 0.270 | <0.001[***] | 0.107 | <0.001[***] |
| UA (μmol/L) | 0.113 | <0.001[***] | 0.185 | <0.001[***] |
| BUN (mmol/L) | −0.077 | 0.002[**] | −0.080 | 0.001[**] |
| hsCRP (mg/L) | −0.028 | 0.538 | −0.103 | 0.024[*] |
| ApoB (g/L) | 0.248 | <0.001[***] | 0.180 | <0.001[***] |
| ApoA (g/L) | 0.001 | 0.968 | −0.311 | <0.001[***] |
| LDL-C (mmol/L) | 0.238 | <0.001[***] | 0.164 | <0.001[***] |
| LVEF | 0.030 | 0.280 | 0.090 | 0.001[**] |

**Notes.**

[*]$P < 0.05$
[**]$P < 0.01$
[***]$P < 0.001$

WBC, White blood cells; SBP, Systolic blood pressure; TC, Total cholesterol; UA, Uric acid; BUN, Blood urea nitrogen; ALB, Albumin; Hb, Hemoglobin; LDL-C, Low-density lipoprotein cholesterol; LVEF, Left ventricular ejected fraction; hsCRP, high sensitivity C reactive protein.

omitting clinically important variables. Univariate logistic regression analysis showed that age (OR = 1.079, 95% CI [1.056−1.102], $P < 0.001$), alcohol history (OR =2.234, 95% CI [1.019−4.896], $P = 0.045$), smoking history (OR = 0.565, 95% CI [0.354−0.902], $P = 0.017$), ALB (OR = 0.847, 95% CI [0.815−0.881], $P < 0.001$), ALT (OR = 1.001, 95% CI [1.001−1.002], $P = 0.001$), BUN (OR =1.135, 95% CI [1.103−1.169], $P < 0.001$), creatinine (OR =1.003, 95% CI [1.002−1.005], $P < 0.001$), Hb (OR =0.970, 95% CI [0.961−0.978], $P < 0.001$), TC (OR =0.991, 95% CI [0.985−0.996], $P = 0.001$), SBP (OR =0.982, 95% CI [0.972−0.992], $P < 0.001$), UA (OR =1.003, 95% CI [1.002−1.004], $P < 0.001$), WBC (OR =1.083, 95% CI [1.043−1.124], $P < 0.001$) and TyG (OR =2.256, 95% CI [1.646−3.092], $P < 0.001$) were significant factors predicting in-hospital death in patients with AMI. The significant variables in the univariate analysis were enrolled in the multivariate logistic regression analysis. We used tolerance Tol and variance inflation factor VIF to evaluate whether the collinearity of included variables was met (Supplemental Materials 4). "Enter" method was used for multivariate logistic regression analysis, and the results showed that age (OR =1.069, 95% CI [1.042−1.097], $P < 0.001$), ALB (OR =0.926, 95% CI [0.879−0.976], $P = 0.004$), ALT (OR =1.001, 95% CI [1.000−1.002],

**Table 3   Logistic regression of predicting in-hospital death in AMI patients without diabetes.**

| Parameters | Univariate regression | | | Variate regression | | |
|---|---|---|---|---|---|---|
| | OR | 95% CI | *P* value | OR | 95% CI | *P* value |
| Age (years) | 1.079 | 1.056–1.102 | <0.001[***] | 1.069 | 1.042–1.097 | <0.001[***] |
| Alcohol | 2.234 | 1.019–4.896 | 0.045[*] | 1.243 | 0.488–3.165 | 0.648 |
| Smoking | 0.565 | 0.354–0.902 | 0.017[*] | 1.097 | 0.607–1.981 | 0.760 |
| ALB (g/L) | 0.847 | 0.815–0.881 | <0.001[***] | 0.926 | 0.879–0.976 | 0.004[**] |
| ALT (U/L) | 1.001 | 1.001–1.002 | 0.001[**] | 1.001 | 1.000–1.002 | 0.028[*] |
| BUN (mmol/L) | 1.135 | 1.103–1.169 | <0.001[***] | 1.063 | 1.005–1.123 | 0.031[*] |
| Creatinine ($\mu$mol/L) | 1.003 | 1.002–1.005 | <0.001[***] | 1.000 | 0.998–1.003 | 0.887 |
| Hb (g/L) | 0.970 | 0.961–0.978 | <0.001[***] | 0.991 | 0.979–1.003 | 0.144 |
| TC (mg/dl) | 0.991 | 0.985–0.996 | 0.001[**] | 0.999 | 0.993–1.005 | 0.698 |
| SBP (mmHg) | 0.982 | 0.972–0.992 | <0.001[***] | 0.982 | 0.971–0.993 | 0.002[**] |
| UA ($\mu$mol/L) | 1.003 | 1.002–1.004 | <0.001[***] | 1.000 | 0.998–1.002 | 0.828 |
| WBC ($10^9$/L) | 1.083 | 1.043–1.124 | <0.001[***] | 1.060 | 1.012–1.111 | 0.014[*] |
| TyG | 2.256 | 1.646–3.092 | <0.001[***] | 3.106 | 2.122–4.547 | <0.001[***] |

Notes.
[*] $P < 0.05$
[**] $P < 0.01$
[***] $P < 0.001$

ALB, Albumin; ALT, Alanine aminotransferase; BUN, Blood urea nitrogen; SBP, Systolic blood pressure; TC, Total cholesterol; UA, Uric acid; WBC, White blood cells.

$P = 0.028$), BUN (OR =1.063, 95% CI [$1.005-1.123$], $P = 0.031$), SBP (OR = 0.982, 95% CI [$0.971-0.993$], $P = 0.002$), WBC (OR =1.060, 95% CI [$1.012-1.111$], $P = 0.014$) and TyG (OR =3.106, 95% CI [$2.122-4.547$], $P < 0.001$), were important factors predicting the prognosis of AMI. "Forward" and "Backward" methods were also performed to validate the result of "enter" method (Supplemental Materials 5). The results of TyG by "Forward" (OR =3.021, 95% CI [$2.084-4.379$], $P < 0.001$) and "Backward" (OR =3.080, 95% CI [$2.123-4.469$], $P < 0.001$) methods were similar with "enter" method. Based on the above results, TyG was an important independent predictor of in-hospital death in AMI after adjusting for multiple clinical confounders (Table 3, Fig. 2A). When TG/HDL-C and significant factors in univariate variables were included in to multivariate analysis, the result of "enter" method showed that TG/HDL-C (OR =1.167, 95% CI [$1.062-1.282$], $P = 0.001$) was also an important predictor of in-hospital death (Table 4, Fig. 2B). The results of TG/HDL-C by "Forward" (OR =1.161, 95% CI [$1.059-1.273$], $P = 0.002$) and "Backward" (OR =1.163, 95% CI [$1.059-1.276$], $P = 0.001$) methods were also similar with "enter" method (Supplemental Materials 5).

## DISCUSSION

This study found that a higher TyG index and TG/HDL-C (markers of insulin resistance) indicated poorer prognosis and higher mortality in AMI patients without diabetes. Higher TyG and TG/HDL-C were also associated with higher inflammatory cells count and LDL-C, which meant they were closely related to severe inflammatory states and dyslipidemia.

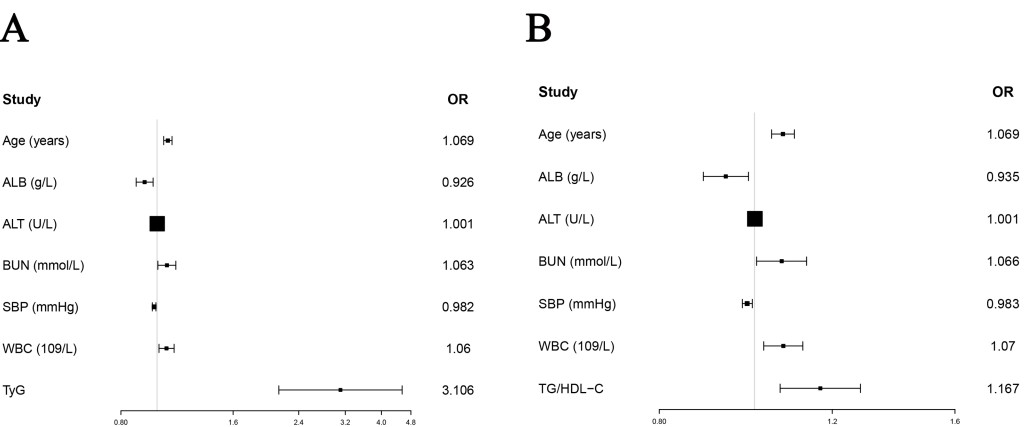

**Figure 2  Forest plots of TyG and TG/HDL-C in predicting in-hospital death in AMI without diabetes.**
(A) The results showed that after adjusting age (OR = 1.069, 95% CI [1.042–1.097], $P < 0.001$), ALB (OR = 0.926, 95% CI [0.879–0.976], $P = 0.004$), ALT (OR = 1.001, 95% CI [1.000–1.002], $P = 0.028$), BUN (OR = 1.063, 95% CI [1.005–1.123], $P = 0.031$), SBP (OR = 0.982, 95% CI [0.971–0.993], $P = 0.002$) and WBC (OR = 1.060, 95% CI [1.012–1.111], $P = 0.014$), TyG (OR = 3.106, 95% CI [2.122–4.547], $P < 0.001$) was an important factor to predict the prognosis of AMI. (B) After adjusting age (OR = 1.069, 95% CI [1.041–1.098], $P < 0.001$), ALB (OR = 0.935, 95% CI [0.887–0.986], $P = 0.011$), ALT (OR = 1.001, 95% CI [1.000–1.002], $P = 0.020$), BUN (OR = 1.066, 95% CI [1.005–1.130], $P = 0.033$), SBP (OR = 0.984, 95% CI [0.972–0.995], $P = 0.005$) and WBC (OR = 1.070, 95% CI [1.022–1.120], $P = 0.004$), TG/HDL-C (OR = 1.167, 95% CI [1.062–1.282], $P = 0.001$) was an important factor to predict the prognosis of AMI.

**Table 4  Logistic regression of predicting in-hospital death in AMI patients without diabetes.**

| Parameters | Univariate regression | | | Variate regression | | |
|---|---|---|---|---|---|---|
| | **HR** | **95% CI** | **P value** | **HR** | **95% CI** | **P value** |
| Age (years) | 1.079 | 1.056–1.102 | <0.001[***] | 1.069 | 1.041–1.098 | <0.001[***] |
| Alcohol | 2.234 | 1.019–4.896 | 0.045[*] | 1.023 | 0.406–2.574 | 0.962 |
| Smoking | 0.565 | 0.354–0.902 | 0.017[*] | 1.101 | 0.611–1.983 | 0.748 |
| ALB (g/L) | 0.847 | 0.815–0.881 | <0.001[***] | 0.935 | 0.887–0.986 | 0.011[*] |
| ALT (U/L) | 1.001 | 1.001–1.002 | 0.001[**] | 1.001 | 1.000–1.002 | 0.020[*] |
| BUN (mmol/L) | 1.135 | 1.103–1.169 | <0.001[***] | 1.066 | 1.005–1.130 | 0.033[*] |
| Creatinine ($\mu$mol/L) | 1.003 | 1.002–1.005 | <0.001[***] | 1.001 | 0.998–1.003 | 0.638 |
| Hb (g/L) | 0.970 | 0.961–0.978 | <0.001[***] | 0.991 | 0.978–1.003 | 0.133 |
| TC (mg/dl) | 0.991 | 0.985–0.996 | 0.001[***] | 1.002 | 0.996–1.008 | 0.564 |
| SBP (mmHg) | 0.982 | 0.972–0.992 | <0.001[***] | 0.983 | 0.972–0.995 | 0.005[**] |
| UA ($\mu$mol/L) | 1.003 | 1.002–1.004 | <0.001[***] | 0.999 | 0.997–1.001 | 0.389 |
| WBC ($10^9$/L) | 1.083 | 1.043–1.124 | <0.001[***] | 1.070 | 1.022–1.120 | 0.004[**] |
| TG/HDL-C | 1.075 | 1.000–1.155 | 0.050 | 1.167 | 1.062–1.282 | 0.001[**] |

**Notes.**
[*]$P < 0.05$
[**]$P < 0.01$
[***]$P < 0.001$
ALB, Albumin; ALT, Alanine aminotransferase; BUN, Blood urea nitrogen; Hb, Hemoglobin; SBP, Systolic blood pressure; TC, Total cholesterol; UA, Uric acid; WBC, White blood cells.

Insulin resistance is characterized by reduced insulin sensitivity and impaired oxidation and glucose utilization (*Wu & Ballantyne, 2020*). Studies have shown that insulin resistance was found not only in type 2 diabetes but also in a wide range of diseases with abnormal physiological functions, triggering the proposal of the concept of insulin resistance syndrome (*Fletcher & Lamendola, 2004*; *Willard, Stevenson & Steenkamp, 2016*). Insulin resistance is associated with the incidence of CVD and could be used as a potential therapeutic target for CVD (*Rutter et al., 2008*; *Bersch-Ferreira et al., 2018*). The gold standard for defining insulin resistance is the euglycemic hyperinsulinemic clamp (*Geloneze & Tambascia, 2006*). The HOMA-IR index is also an effective marker for diagnosing insulin resistance (*Geloneze & Tambascia, 2006*). However, both of them are costly, and the identification of HOMA-IR is based on fasting insulin, which is not readily available in a large-scale cross-sectional study or a retrospective design. A study that used TyG and TG/HDL-C index as surrogate markers of insulin resistance to explore the longitudinal association of insulin resistance with the progression of arteriosclerosis in the hypertensive population found that the synergistic effect of hypertension and insulin resistance led to the adverse progression of arteriosclerosis (*Wu et al., 2021*). Therefore, TyG and TG/HDL-C are expected to perform as reliable and convenient markers, not inferior to the above two indicators.

Based on TyG and TG/HDL-C, we found insulin resistance was associated with inflammatory states and dyslipidemia. Previous studies have demonstrated that the onset age of stroke and hypertension decreased gradually with the increase of TyG index, which clarifies the negative effects of TyG index on the onset of diseases (*Zhou et al., 2020*; *Gu et al., 2020*). Insulin resistance is always accompanied by chronic low-grade inflammation (*Perry et al., 2021*). Increased WBC counts, neutrophil counts and lymphocyte counts are convenient markers for evaluating systemic inflammation (*Perry et al., 2021*). In our study, the WBC counts and lymphocyte counts were significantly associated with TyG index and TG/HDL-C, indicating that insulin resistance may result from systemic inflammation. A higher TyG index was accompanied by more severe inflammation. Moreover, the relationship between dyslipidemia and insulin resistance was reciprocal (*Fassler et al., 2019*). On the one hand, lipid accumulation is one of the leading causes of insulin resistance (*Wang et al., 2021a*; *Wang et al., 2021b*). On the other hand, insulin resistance is related to changes in lipoprotein and lipid metabolism (*van der Kolk et al., 2019*). We found that TC and LDL-C levels were positively associated with higher TyG and TG/HDL-C, which confirmed that insulin resistance was accompanied by dyslipidemia and other metabolic disorders.

Insulin resistance is not only important in AMI patients with diabetes, but also in patients without diabetes. Many clinical trials have determined that insulin resistance plays a vital role in the development of CVD and can result in myocardial ischemia (*Klarin et al., 2017*; *Masaki et al., 2019*). Furthermore, insulin resistance is currently considered an independent risk factor for cardiovascular death (*Zhang et al., 2021*). In a large, multi-ethnic and prospective cohort study of more than 15,000 postmenopausal women published in Circulation: Cardiovascular Quality and Outcomes, insulin levels, HOMA-IR, blood glucose, and TG/HDL-C were measured to assess insulin resistance. After adjusting for

traditional cardiovascular risk factors such as age, race/ethnicity, smoking, total cholesterol, systolic blood pressure, and treatment with antihypertensive agents, insulin resistance was significantly associated with increased CVD risk (*Schmiegelow et al., 2015*). Previous studies on TyG and TG/HDL-C and insulin resistance mainly focused on AMI patients with diabetes (*Zhang et al., 2021*). However, we mainly explored the role of TyG and TG/HDL-C in predicting the prognosis of AMI patients without diagnosis of diabetes due to diabetes being an significant risk factor for AMI. It was interesting and significant for us to find insulin resistance can help predict short-term death in non-diabetes AMI patients.

There are still some limitations of our study. First, the long-term follow-up was not performed, limiting the survival analysis of high TyG and TG/HDL-C patients. Second, the roles of TyG and TG/HDL-C in AMI patients with or without diabetes were not compared. Third, HOMA-IR and serum insulin levels were not measured.

## CONCLUSIONS

In conclusion, the emerged simple markers of insulin resistance, TyG index and TG/HDL-C are important predictors of in-hospital death in AMI patients without diabetes after adjusting significant variables such as age, SBP, BUN, WBC, ALT and ALB. For the population without diabetes, it is also worth emphasizing the importance of screening for insulin resistance and actively taking early intervention measures.

## ACKNOWLEDGEMENTS

None.

### Funding

This study was funded by the National Natural Science Foundation of China [granted number NSFC 81770452 and 82070295]. The funders had no role in study design, data collection and analysis, decision to publish, or preparation of the manuscript.

### Grant Disclosures

The following grant information was disclosed by the authors:
the National Natural Science Foundation of China: 81770452, 82070295.

### Competing Interests

The authors declare there are no competing interests.

### Author Contributions

- Jiaqi Guo conceived and designed the experiments, performed the experiments, analyzed the data, prepared figures and/or tables, authored or reviewed drafts of the article, and approved the final draft.
- Zhenjun Ji conceived and designed the experiments, performed the experiments, analyzed the data, prepared figures and/or tables, authored or reviewed drafts of the article, and approved the final draft.
- Abdlay Carvalho performed the experiments, authored or reviewed drafts of the article, and approved the final draft.
- Linglin Qian performed the experiments, authored or reviewed drafts of the article, and approved the final draft.
- Jingjing Ji performed the experiments, authored or reviewed drafts of the article, and approved the final draft.
- Yu Jiang performed the experiments, authored or reviewed drafts of the article, and approved the final draft.
- Guiren Liu performed the experiments, analyzed the data, authored or reviewed drafts of the article, and approved the final draft.
- Genshan Ma conceived and designed the experiments, authored or reviewed drafts of the article, and approved the final draft.
- Yuyu Yao conceived and designed the experiments, authored or reviewed drafts of the article, and approved the final draft.

## Human Ethics

The following information was supplied relating to ethical approvals (i.e., approving body and any reference numbers):

This study was approved by the Ethics Committee of Zhongda Hospital affiliated to Southeast University (2020ZDSYLL164-P01).

## Data Availability

The raw data is available in the Supplemental File.

## Supplemental Information

Supplemental information for this article can be found online at http://dx.doi.org/10.7717/peerj.14346#supplemental-information.

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
