# Peer review of "The triglycerides-glucose index and the triglycerides to high-density lipoprotein cholesterol ratio are both effective predictors of in-hospital death in non-diabetic patients with AMI"

_PeerJ, doi:10.7717/peerj.14346_

## Round 0.1 · original submission · Major Revisions

Thank you for your submission. Based on the reviewers' comments, I suggest major revisions for this manuscript. Please respond to reviewers' comments, especially regarding statistical analysis.

·

Basic reporting

The manuscript is written concisely, but also comes along with minor grammatical errors in some sentences. All the literature references on this paper lack numbering which causes difficulties in tracking.

In the abstract section, the acronyms that appear for the first time should come along with the full description (i.e. line 32). And it would be better to have a clear layout for the result and conclusion part in this section.

The introduction section cited several previous research results on ACS, insulin resistance, TyG, and TG/HDL-C in a very detailed way. But the order and structure of each paragraph make the relationship explanation ambiguous.

In the Materials & Methods section, itemized list with comparison is suggested for population description, and the data collection part should consider using a table that is more readable. The statistics part lacks detailed indication and explanation such as which data follows a normal distribution, why T-test, why Pearson analysis, etc. The experiment group detail is insufficient.

The result section is suggested to have the corresponding statistic analysis table under each subsection, listing all the P-value and confidence intervals in one paragraph causes an indecipherable reading experience.

A self-contained result and relevant studies are discussed in the Discussion section.

Experimental design

The experiment part needs more details, such as the criteria for grouping patients, and whether other factors other than TyG and TG/HDL-C would affect the statistical analysis result. Also, explain the main reason for choosing those particular statistical analysis methods (T-test, ANOVA, logistic regression), and the way to handle outliers in raw data (if any).
Sufficient details for experimental design are highly suggested.

Validity of the findings

Controlled and clear statistical analysis is required for delivering the findings.

Additional comments

A clear layout of each subsection is recommended. In addition to just listing statistical values, it's worthwhile to document those significant factors and have an extra discussion.

Reviewer 2 ·

Basic reporting

Overall, the authors did a good job laying out their research motivation, data collection and discoveries, with the exception on the data analysis part. The writing is clear and easy to understand. The supportive tables of data are relevant, and the captions are proper that provide enough info to support the findings.

On the clinical background side, the authors demonstrate strong knowledge on the prognosis in non-diabetic acute myocardial infarction (AMI) patients. Lengthy and sufficient literature review is provided in both intro and conclusion sections in the paper.

On the data analysis side, the authors opted for the very conventional statistical analysis methods, Pearson correlation, t-tests and logistic regression. Judging from the purpose of this particular research and the data type, all of these 3 methods are reasonable. If done properly, all the methods should be able to provide sound and valid findings. However, the authors failed to convince me that they had done such a proper work. Largely, there are essential steps of the data analysis missing from the paper. Please see next comment section for more details.

Experimental design

Overall, the authors provided rigorous and well thought-after design of experiment. The patient selection process is reasonable and serves the purpose of the paper.

Validity of the findings

T-tests and logistic regressions are relatively easy approaches to detect significant differences of covariates between a control group and treatment group (death and no-death groups in this particular paper).

I am happy to see that the authors chose the right t-test, independent t-test, as opposed to paired t-test. However, within independent t-test method, there are 2 different approaches, one for equal variance between groups, one for unequal variances between groups. The authors did not provide data to support one approach over the other. The authors stated that “For the data conforming to a normal distribution, independent-sample T-test was used”, which makes me wonder if they did a normal transformation on the data before t-test, and if so, which normal transformation is done?

For the logistic analysis, the authors did not run a collinearity test on the covariates before running the multivariate logistic regression, which could lead to biased estimation of their concluded predictive effect. The authors also did not explain why they wanted to run a univariate regression to choose significant variables that would be included in the multivariate regression. The insignificant univariates could still be significant when presented with other covariates. The absence of these steps casts doubt on their findings. I encourage the authors to add these steps for revision.

Another issue is that “adjusting multiple clinical confounders” and “After adjusting for most cardiovascular risk factors” are not clear to me how the authors actually did the adjustments in the data analysis.

Reviewer 3 ·

Basic reporting

1. The paper is generally well written. The background and literature review are carefully written. The tables and figures are generally well labeled and described, while more detail should be provided in the notes or main texts.
2. There are some tiny grammar issues.
3. The raw data seems incomplete, for example, the death information is missing

Experimental design

1. The important outcome the in-hospital death in the research question is not clearly defined, how this research fills an identified knowledge gap should be emphasized, and the scientific question should be clearly defined without any ambiguity
2. The analysis methods do not have sufficient detail and justification

Validity of the findings

1. Could you justify the assumptions for the use of t-test, and Pearson correlation, and clarify what tests are you using for the p-values of Pearson correlation
2. All-cause mortality is usually treated as a survival outcome. Could you clearly define the in-hospital death and justify the choice of the logistic model instead of the survival model.
3. The model-building procedure and interpretation are incorrect. A proper model selection procedure should be selected. The model should be correctly interpreted.

---

## Round 0.2 · Minor Revisions

Thank you for submitting the manuscript. Please address the remaining minor comments from the reviewers.

·

Basic reporting

1. The revised manuscript is well written, and the background and analytic method are written in detail. References are cited properly.
2. Different statistical methods are applied to analyze the factor significance, and the methodologies are interpreted in a detailed way.
3. The paper structure and layout are clear.

Experimental design

1. The statistical methods are discussed carefully, and the scientific questions are answered well.
2. Factors are analyzed carefully with proper statistical methods. The author did a good job in the Result section by listing and discussing each subsection gradually.

Validity of the findings

1. Statistical results are well discussed and interpreted, and the conclusions are well stated.

Reviewer 2 ·

Basic reporting

As stated in the previous review, the paper read well and easy to understand.

Experimental design

Same like the basic reporting, there is no need to change the experiment design. The authors did a good job in both the previous version and the new version.

Validity of the findings

The major criticism was on the statistical analysis in the original version of the paper. It is great to see that the authors addressed those feedbacks and added more explanations in this new version. The homogeneity test on the variances across groups has beed added. The reasoning behind the model selection (inclusions of univariates) has been added. And the normality of the data is tested. Overall, the improvement is sufficient to pass the peerj standard.

Reviewer 3 ·

Basic reporting

Thanks for the revision. I don't have furthur comments and concerns.

Experimental design

Thanks for the changes and clarification. I don't have furthur comments or concerns.

Validity of the findings

Thanks for the revision. I still have some concerns with respect to the multivariate logistic model and interpretation.

1. The authors only included significant variables in the univariate analysis to the multivariate logistic regression analysis, which is very strict criteria and may miss important variables for the multivariate model.
2. The author did not do further model selection for the multivariate model. Could you give more explanation or justification?
3. In the conclusion, I think it is better to make it clear that “TyG index and TG/HDL-C are important predictors of in-hospital death in AMI patients without diabetes” adjusting for other important covariates.

---

## Round 0.3 · accepted · Accept

Thank you for addressing all reviewers' comments.

Reviewer 3 ·

Basic reporting

Thanks for the revision, I don't have furthur comments/questions.

Experimental design

Thanks for the revision, I don't have furthur comments/questions.

Validity of the findings

Thanks for the revision, I don't have furthur comments/questions.